# On-surface synthesis of a doubly anti-aromatic carbon allotrope

Yueze Gao[1,5], Florian Albrecht[2,5], Igor Rončević[1,3], Isaac Ettedgui[1], Paramveer Kumar[1], Lorel M. Scriven[1], Kirsten E. Christensen[1], Shantanu Mishra[2], Luca Righetti[4], Max Rossmannek[4], Ivano Tavernelli[4], Harry L. Anderson[1✉] & Leo Gross[2✉]

Synthetic carbon allotropes such as graphene[1], carbon nanotubes[2] and fullerenes[3] have revolutionized materials science and led to new technologies. Many hypothetical carbon allotropes have been discussed[4], but few have been studied experimentally. Recently, unconventional synthetic strategies such as dynamic covalent chemistry[5] and on-surface synthesis[6] have been used to create new forms of carbon, including γ-graphyne[7], fullerene polymers[8], biphenylene networks[9] and cyclocarbons[10,11]. Cyclo[$N$]carbons are molecular rings consisting of $N$ carbon atoms[12,13]; the three that have been reported to date ($N = 10$, 14 and 18)[10,11] are doubly aromatic, which prompts the question: is it possible to prepare doubly anti-aromatic versions? Here we report the synthesis and characterization of an anti-aromatic carbon allotrope, cyclo[16]carbon, by using tip-induced on-surface chemistry[6]. In addition to structural information from atomic force microscopy, we probed its electronic structure by recording orbital density maps[14] with scanning tunnelling microscopy. The observation of bond-length alternation in cyclo[16]carbon confirms its double anti-aromaticity, in concordance with theory. The simple structure of $C_{16}$ renders it an interesting model system for studying the limits of aromaticity, and its high reactivity makes it a promising precursor to novel carbon allotropes[15].

Many cyclo[$N$]carbons ($N = 6$–40) have been detected in the gas phase[12,13,16], and two examples ($C_6$ and $C_8$) have been trapped in solid argon and characterized by infrared spectroscopy[17,18]. Cyclo[10]carbon, cyclo[14]carbon and cyclo[18]carbon have been characterized by scanning probe microscopy of individual molecules on NaCl surfaces at low temperature[10,11,19]. Atomic force microscopy (AFM) images revealed cumulenic structures for $C_{10}$ and $C_{14}$ with bond-angle alternation (BAA)[11] and a polyynic structure for $C_{18}$ (refs. 10,19). Cyclo[$N$]carbons with $N = 4n + 2$ (where $n$ is an integer), such as $C_{10}$, $C_{14}$ and $C_{18}$, are expected to be doubly aromatic and to have special stability, due to their closed-shell electronic configurations, relating to the presence of in-plane and out-of-plane aromatic Hückel circuits of $4n + 2$ π electrons[20–25]. By contrast, cyclo[$4n$]carbons have been predicted to be less stable and doubly anti-aromatic[22–26]. Here we report the first structural characterization of a cyclo[$4n$]carbon to our knowledge. $C_{16}$ was prepared on a NaCl surface by tip-induced chemistry from a $C_{16}(CO)_4Br_2$ precursor. AFM and scanning tunnelling microscopy (STM) provide insight into the geometry and electronic structure, respectively, of neutral $C_{16}$ and anionic $C_{16}^-$. We find that neutral $C_{16}$ exhibits significant bond-length alternation (BLA), which confirms its double anti-aromaticity. Our experimental results are complemented by state-of-the-art quantum mechanical calculations, as well as by methods suitable for execution on a quantum computer.

Cyclocarbons have two orthogonal π systems, one with orbital lobes in the ring plane and the other out of plane, with nodes in the ring plane. In an infinitely large cyclocarbon, these two π systems are degenerate; but in a finite ring, in-plane frontier orbitals are slightly higher in energy than their out-of-plane counterparts[13]. This pattern of orbitals can lead to several possible electronic states. In the $D_{16h}$ geometry of $C_{16}$ with no BLA, the ground state may be a doubly aromatic |2200> state (Fig. 1, left), with 18 ($4n + 2$) and 14 ($4n - 2$) electrons in out-of-plane and in-plane π systems, respectively. In this state, there are two degenerate pairs of frontier orbitals (out-of-plane A'' and B'' are occupied, and in-plane A' and B' are unoccupied). If we introduce BLA ($D_{8h}$ symmetry), these orbital pairs cease to be degenerate, with one member of each pair (A in Fig. 1, right) becoming stabilized relative to the other (B). This symmetry breaking leads to a doubly anti-aromatic |2020> configuration with 16 electrons in both in-plane and out-of-plane π systems. A third possible state would be |1111> , with $D_{16h}$ symmetry, but such open-shell configurations are known to be unstable relative to closed-shell alternatives[27].

The unique structure, small size and high symmetry of cyclocarbons has made them a target of many theoretical studies, sometimes producing conflicting results[13]. Here, we investigate $C_{16}$ using both state-of-the-art computational methods and a variational quantum eigensolver[28] paired with the quantum unitary coupled-cluster singles

[1]Department of Chemistry, Oxford University, Chemistry Research Laboratory, Oxford, UK. [2]IBM Research Europe – Zürich, Rüschlikon, Switzerland. [3]Institute of Organic Chemistry and Biochemistry of the Czech Academy of Sciences, Prague, Czechia. [4]IBM Quantum, IBM Research – Zürich, Rüschlikon, Switzerland. [5]These authors contributed equally: Yueze Gao, Florian Albrecht. ✉e-mail: harry.anderson@chem.ox.ac.uk; lgr@zurich.ibm.com

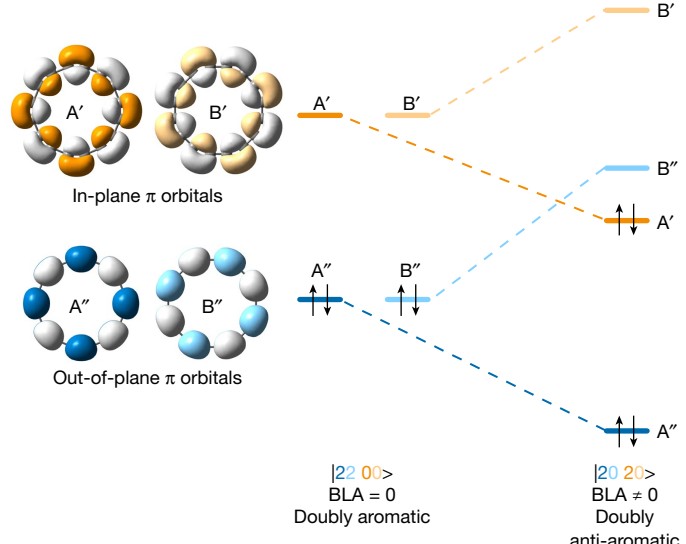

**Fig. 1 | Frontier orbitals of two electronic states of $C_{16}$.** In-plane orbitals are labelled A′ and B′ and out-of-plane orbitals A″ and B″. Orbitals A′, B′ (and A″, B″) are related by rotation and have equal energy when all bonds are of equal length. Introducing BLA lifts this degeneracy, resulting in orbital reordering and a doubly anti-aromatic ground state.

**Fig. 2 | Synthesis of $C_{16}$.** TMEDA is tetramethylethylenediamine.

and doubles (q-UCCSD)[29] ansatz. These calculations confirm that the doubly anti-aromatic configuration is the ground state of $C_{16}$, with strong BLA.

## Precursor synthesis

Cyclo[16]carbon was synthesized as shown in Fig. 2. Glaser–Hay coupling of a mixture of alkynes **1** and **2** gave macrocycle **3** in 20% yield, and the structure of this product was confirmed by single-crystal X-ray diffraction (Supplementary Fig. 3). Compounds **3** and **4** are anti-aromatic (as confirmed by the [1]H nuclear magnetic resonance spectrum of compound **3**; Supplementary Fig. 2). Deprotection of **3** to give **4** proved difficult because of the high reactivity of compound **4**, but after testing many reaction conditions, we found that **3** can be converted to **4** in 94% yield using trifluoroacetic acid containing water (2.5% by volume).

## On-surface synthesis and characterization

Precursor **4** was sublimed by fast heating from a Si wafer[10] onto a Cu(111) single-crystal surface partially covered with NaCl at a sample temperature of about $T = 10$ K. On-surface synthesis (Fig. 3a) and characterization by STM and AFM with CO-tip functionalization[30,31] were performed at $T = 5$ K. We found intact molecules of **4** on bilayer NaCl, denoted NaCl(2 ML)/Cu(111), as shown in Fig. 3b. The Br atoms appear as bright (repulsive) dots in the AFM image[19], whereas the CO masking groups are dark features[10]. The triple bonds show up as bright features due to bond-order related contrast obtained with CO-tip functionalization[10,31,32] (for further data on **4**, see Supplementary Fig. 6).

Voltage pulses applied for a few seconds at constant tip height were used to unmask the acetylenes in individual molecules of precursor **4**. We successively increased the voltage and decreased the tip height for the pulse until it resulted in dissociation reactions. For tunnelling currents on the order of few pA, the minimum voltage required for debromination of **4** to give **5** (Fig. 3c; see Supplementary Fig. 7 for further data) was 1.3 V, coinciding with the bias for resonant tunnelling: that is, electron attachment to **4** (Supplementary Fig. 6). For CO unmasking, larger bias voltages were required, typically about 3 V. We speculate that the dissociation reactions are triggered in transiently charged species by inelastic electron tunnelling processes[31]. Intermediate **6** was

observed after dissociating the first pair of CO masking groups (Fig. 3d; see Supplementary Fig. 8 for further data). Removal of a second pair of CO molecules gave the final product, $C_{16}$ (Fig. 3e and Supplementary Figs. 9 and 10). Previously, gas-phase $C_{16}$ has been formed from a molecular precursor[33,34] and studied in its anionic[34,35] and cationic[16,36] forms, but to our knowledge, this is the first time $C_{16}$ has been generated in a condensed phase or structurally characterized. The yield for the on-surface synthesis of $C_{16}$ was about 30%; in unsuccessful attempts, the ring opened to form linear polyynic chains (Supplementary Fig. 11) or the molecule was picked up by the tip.

We observed $C_{16}$ in two different forms on the NaCl surface (Fig. 3f,g) that we assign to neutral $C_{16}^{0}$ and negatively charged $C_{16}^{-}$, respectively (see also Fig. 4, Supplementary Figs. 12 and 13 and Supplementary Tables 1 and 2). Whereas $C_{16}^{0}$ appears circular, $C_{16}^{-}$ adopts a distorted oval geometry. We observed a variety of adsorption sites for $C_{16}^{0}$ on the NaCl surface (Supplementary Fig. 14), indicating a weak interaction with the substrate. In contrast, $C_{16}^{-}$ showed a systematic preference for adsorption above a bridge site (Supplementary Figs. 15 and 16). To investigate the interaction of $C_{16}^{0}$ and $C_{16}^{-}$ with the NaCl surface, we performed density functional theory (DFT) calculations with periodic boundary conditions, both on a pristine surface and at NaCl island step edges. The calculated lowest-energy adsorption sites of $C_{16}^{0}$ and $C_{16}^{-}$ on pristine NaCl are shown in Fig. 3h,i, respectively. For the neutral charge state, we calculated an adsorption energy of 0.65 eV, similar to the value of 0.67 eV previously calculated for $C_{18}$ on NaCl (ref. 37) that was predicted to diffuse freely across the surface even at low temperatures. The calculated relaxed adsorption geometry of $C_{16}^{-}$ on pristine NaCl is oval shaped, with the molecule centred on a bridge site (Fig. 3i), in agreement with its experimentally observed site and shape (Fig. 3g, Supplementary Fig. 15 and Supplementary Table 1). This adsorption geometry can be attributed to electrostatic interactions of the $C_{16}^{-}$ anion with the Na cations and Cl anions, resulting in a substantially stronger adsorption energy (1.44 eV) than that of the neutral molecule.

The $C_{16}$ molecules frequently moved on the surface during imaging with AFM and STM, indicating a small diffusion barrier and making them challenging to characterize. We never observed neutral $C_{16}$ stably isolated on the NaCl surface, but always near a third-layer NaCl step edge, to provide a more stable adsorption site and facilitate detailed characterization. Figure 3j–m shows $C_{16}$ adsorbed in a bay of a third-layer island imaged with AFM at different tip heights. Kelvin probe force spectroscopy confirmed that the molecule in Fig. 3f,j–m is charge neutral (Supplementary Tables 2 and 3 and Supplementary Fig. 17). The bright contrast obtained by CO-tip AFM above the triple

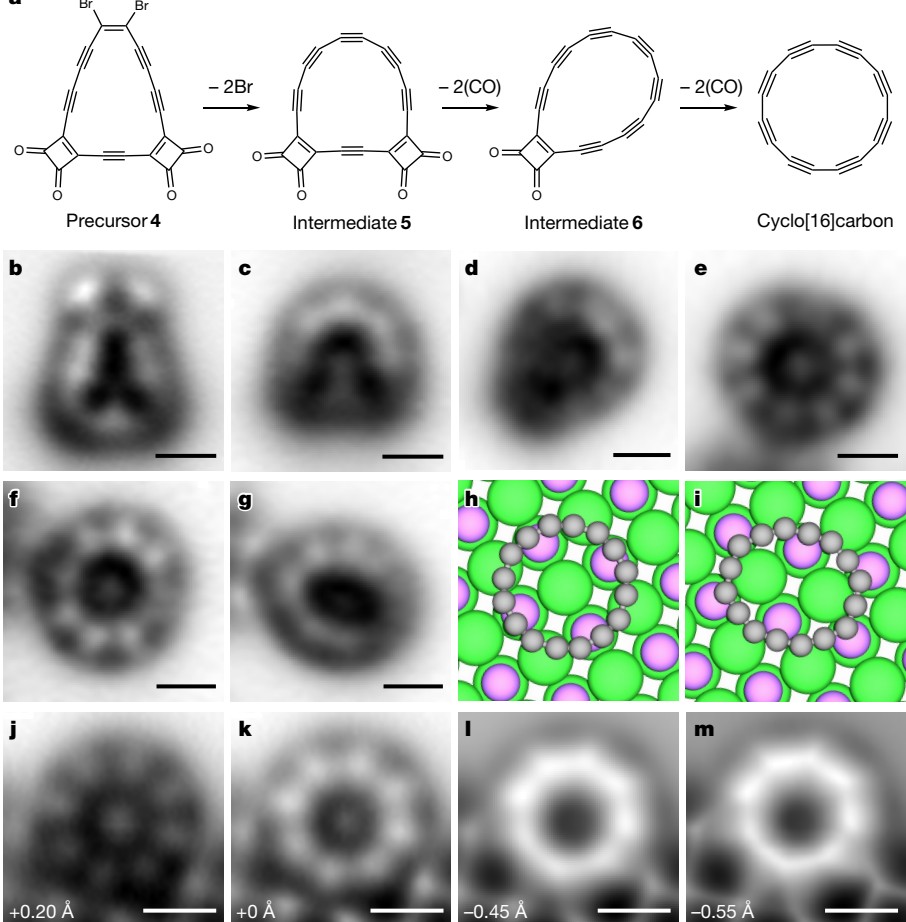

**Fig. 3 | On-surface synthesis of C₁₆ and structural characterization.**
**a**, Reaction scheme. **b**–**e**, Constant-height, CO-tip AFM images of precursor **4**
(**b**), intermediates **5** (**c**) and **6** (**d**), and C₁₆ (**e**). **f**–**i**, AFM image of C₁₆ in neutral (**f**)
and anionic (**g**) charge state, and calculated lowest-energy adsorption sites of
C₁₆⁰ (**h**) and C₁₆⁻ (**i**) on NaCl (colour code: Na pink, Cl green). **j**–**m**, C₁₆⁰ adsorbed

in a bay of a third-layer NaCl island, imaged with AFM at different decreasing
tip-height offsets: +0.20 Å (**j**), +0 Å (**k**), −0.45 Å (**l**) and −0.55 Å (**m**). All molecules
are adsorbed on NaCl(2 ML)/Cu(111). The tip-height offsets provided in the
images refer to the STM setpoint of $I$ = 0.2 pA and $V$ = 0.2 V on bare NaCl(2 ML)/
Cu(111). Scale bars, 0.5 nm.

bonds for larger tip heights (Fig. 3j,k) evolves to the shape of an octagon
with corners at the positions of triple bonds at decreased tip heights
(Fig. 3l,m). The results indicate BLA[10]: that is, a polyynic structure of
neutral C₁₆. Our computations (Supplementary Table 1 and Supple-
mentary Figs. 54 and 55) predict a larger adsorption energy (1.13 and
2.61 eV for C₁₆⁰ and C₁₆⁻, respectively) at defect sites compared to the
pristine surface, accompanied by an increase in BAA (up to 35° for
C₁₆⁰ and up to 50° for C₁₆⁻, compared to 20–30° on a pristine surface).
BLA is maintained in all cases, with no fundamental changes in the
electronic structure.

## Charge-state switching

The charge state of C₁₆ can be controllably switched using the applied
bias, as shown in Fig. 4. At about $V$ = 0.5 V, the molecule switched from
neutral C₁₆⁰ to the anion C₁₆⁻, (and at $V$ = −0.3 V in the reverse direc-
tion, C₁₆⁻ to C₁₆⁰; Supplementary Figs. 12 and 18). The STM images in
Fig. 4a,b show C₁₆⁰ and C₁₆⁻, respectively. The negative charge state
leads to a characteristic dark halo (Fig. 4b) and interface state scat-
tering as observed in the difference image Fig. 4c (ref. 38); see also
Supplementary Fig. 12 for images with enhanced contrast. The assign-
ments of these charge states are corroborated by Kelvin probe force
spectroscopy (Supplementary Fig. 18). AFM data for C₁₆⁰ and C₁₆⁻ are
shown in Fig. 4d,e with corresponding Laplace-filtered data in Fig. 4g,h,
respectively. In this case, the structural distortion of C₁₆⁰ and C₁₆⁻ is

similar, which we assign to the influence of the third-layer NaCl island
(Supplementary Table 1).

The more stable adsorption at the third-layer island allowed us to
image the molecule at increased bias voltages without inducing move-
ment of the molecule. At about 1.2 V, we observe the onset of an elec-
tronic resonance by scanning tunnelling spectroscopy (Supplementary
Fig. 18). The STM image at 1.2 V shown in Fig. 4f (Laplace-filtered data in
Fig. 4i), reveals the orbital density corresponding to that resonance[14].
As the molecule is already in the anionic charge state at $V$ > 0.5 V, we
assign this resonance to the transition from anionic C₁₆⁻ to the dianionic
charge state C₁₆²⁻, giving us insight into the electronic structure of C₁₆.

Multireference methods and DFT (see Supplementary Tables 4 and
5 for details) both predict C₁₆⁰ to have a |2020⟩ ground state with a
polyynic geometry and BLA, but no BAA, in the gas phase ($D_{8h}$ symme-
try, Fig. 4k). The electronic structure of C₁₆⁰ (Fig. 1, right, and Supple-
mentary Fig. 19) features a nearly degenerate pair of highest occupied
molecular orbitals A″ (HOMO−1) and A′ (HOMO), as well a nearly degen-
erate pair of lowest unoccupied molecular orbitals B″ (LUMO) and
B′ (LUMO + 1). A′ and B′ (as well as A″ and B″) are related by rotation; in
a magnetic field, they couple to induce a strong ring current (−25 nA T⁻¹,
cf. 12 nA T⁻¹ in benzene), reinforcing the applied field inside the ring.
This current is a signature of anti-aromaticity[39], and it can be visualized
by nucleus-independent chemical shift calculations (see comparison
of the plots for the |2200⟩ and |2020⟩ states of C₁₆ in Supplementary
Fig. 28). In contrast to the neutral |2020⟩ state of C₁₆, the anion shows

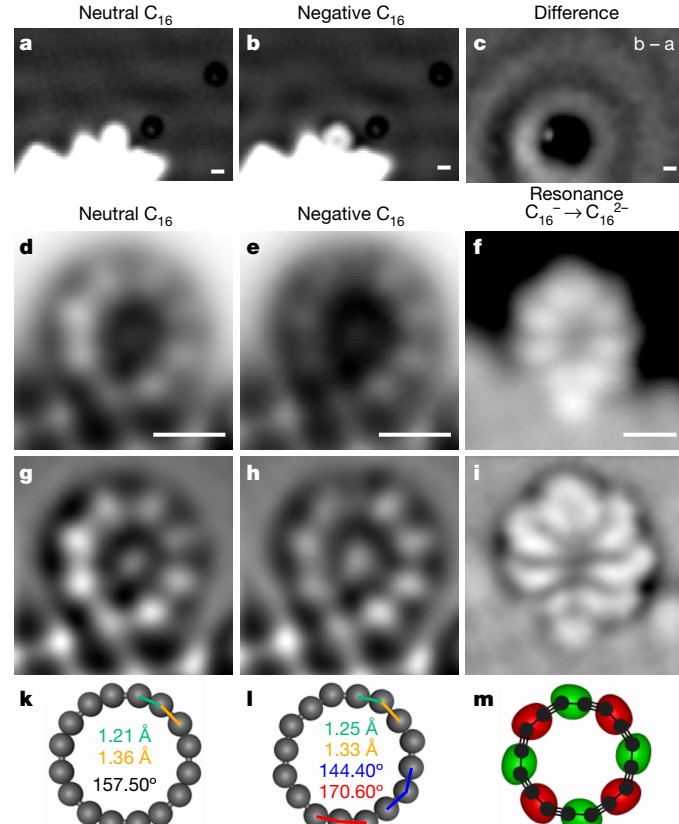

Neutral C$_{16}$ | Negative C$_{16}$ | Difference

**a** | **b** | **c** (b − a)

Neutral C$_{16}$ | Negative C$_{16}$ | Resonance C$_{16}^-$ → C$_{16}^{2-}$

**d** | **e** | **f**

**g** | **h** | **i**

**k**
1.21 Å
1.36 Å
157.50°

**l**
1.25 Å
1.33 Å
144.40°
170.60°

**m**

**Fig. 4 | Charge-state switching and electronic characterization.**
**a,b**, Constant-current STM images of C$_{16}$ in neutral (**a**) and negative charge state (**b**), respectively ($V$ = 50 mV, $I$ = 0.2 pA). **c**, Difference of panels **b** and **a**. **d,e**, Constant-height AFM images of C$_{16}^0$ (**d**) and C$_{16}^-$ (**e**). **f**, Constant-current STM ($I$ = 0.4 pA and $V$ = +1.2 V) mapping the ionic resonance of C$_{16}^-$ to C$_{16}^{2-}$. **g–i**, Same data as **d–f** after applying a Laplace filter. The molecule was adsorbed on NaCl(2 ML)/Cu(111) near a third-layer island. Scale bars, 0.5 nm. **k,l**, Optimized geometries (ωB97XD/def2-TZVP) of C$_{16}^0$ (**k**) and C$_{16}^-$ (**l**), with bond lengths and bond angles indicated. **m**, Simulated isosurface at 0.2 atomic units (1.4 e/Å$^{-3}$) of the LUMO of C$_{16}^-$.

both BLA and BAA (Fig. 4l), due to single occupation of the B″ orbital, resulting in $C_{8h}$ symmetry.

The DFT-predicted LUMO of C$_{16}^-$ (Fig. 4m and Supplementary Fig. 20) can be compared to the electronic resonance imaged by STM (Fig. 4f,i), which corresponds to the squared orbital wavefunction[14,38], and to the addition of a second electron to the singly occupied out-of-plane orbital (B″) in C$_{16}^-$. Both theory and experiment show high-density lobes above the long bonds of C$_{16}^-$, which are located between the bright features of the corresponding AFM images. The symmetry lowering from $D_{8h}$ to $C_{8h}$ in C$_{16}^-$, which is the effect of BAA, is reflected in the shape of the orbital lobes and can be observed in both experiment (Fig. 4f,i) and theory (Fig. 4m). AFM data showing BLA, and STM data showing the orbital density for the C$_{16}^-$ to C$_{16}^{2-}$ transition, corresponding to the addition of an electron to the B″ orbital of C$_{16}^-$, are all in excellent agreement with the calculations, strongly indicating the doubly anti-aromatic character of C$_{16}^0$, which causes pronounced BLA and a $D_{8h}$ geometry. The two other possible electronic configurations of C$_{16}$, doubly aromatic |2200⟩ and open-shell |1111⟩, were calculated by DFT to have nearly identical $D_{16h}$ minima with no BLA and substantially higher energies (2.47 and 1.78 eV, respectively) than the doubly anti-aromatic |2020⟩ ground state. Relative ground-state energies of the $D_{8h}$ and $D_{16h}$ minima were also determined using q-UCCSD by simulating quantum circuits with Qiskit[40]. q-UCCSD predicts that the $D_{8h}$ minimum is more stable than the $D_{16h}$ minimum by 3.38 eV, which is very similar to the result obtained

using conventional coupled-cluster singles and doubles (3.31 eV; see Supplementary Information for further discussion).

Our experimental results, most importantly the observed BLA for neutral C$_{16}$, confirm the occupation of both π systems (in-plane and out-of-plane) with 16 electrons, making the molecule doubly anti-aromatic. Ring current calculations on neutral C$_{16}$ also indicate significant anti-aromaticity in this electronic configuration. The investigation of both C$_{16}^0$ and C$_{16}^-$ provides confidence in the assignment of charge states and insights into the electronic structure of the molecule. The synthesis, stabilization and characterization of C$_{16}$ opens the way to create other elusive carbon-rich anti-aromatic molecules by atom manipulation.

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

# Article

## Data availability

The data that support the findings of this study are available in the paper and its Supplementary Information, or are available from the Zenodo public repository (https://zenodo.org/record/8226451 and https://doi.org/10.5281/zenodo.8226451). Crystallographic data for compound **3** are available free of charge from the Cambridge Crystallographic Data Centre (CCDC 2240722), https://www.ccdc.cam.ac.uk/data_request/cif.

**Acknowledgements** We thank the following organizations for support: European Research Council grant no. 885606, ARO-MAT (H.L.A., Y.G.); European Community Horizon 2020 grant project 101019310 CycloCarbonCatenane (Y.G., H.L.A.); European Community grant ElDelPath (I.R., H.L.A.); Leverhulme Trust (Project Grant RPG-2017-032) (H.L.A., L.M.S.); European Research Council Synergy grant MolDAM (grant no. 951519); and European Union project SPRING (grant no. 863098). Computational resources were provided by Cirrus UK National Tier-2 HPC Service at EPCC (http://www.cirrus.ac.uk), funded by the University of Edinburgh and EPSRC (EP/P020267/1); and the Ministry of Education, Youth and Sports of the Czech Republic through the e-INFRA CZ (ID:90140). IBM, the IBM logo, and ibm.com are trademarks of the International Business Machines Corp., registered in many jurisdictions worldwide. Other product and service names might be trademarks of IBM or other companies. The current list of IBM trademarks is available at https://www.ibm.com/legal/copytrade.

**Author contributions** H.L.A. and L.G. conceived and initiated the project. Y.G., I.E., P.K. and L.M.S. synthesized the precursors. F.A. and L.G. carried out the atom manipulation and scanning probe microscopy. I.R., L.R., M.R. and I.T. performed theoretical analysis and computational simulations. K.E.C. determined the crystal structure of compound **3**. Y.G., F.A., I.R., H.L.A. and L.G. wrote the paper. All authors discussed the results and edited the manuscript.

**Competing interests** The authors declare no competing interests.

## Additional information

**Correspondence and requests for materials** should be addressed to Harry L. Anderson or Leo Gross.

