## [Peer Review File · Nature]

Manuscript Title: On-Surface Synthesis of a Doubly Anti-Aromatic Carbon Allotrope

Reviewer Comments & Author Rebuttals

Reviewer Reports on the Initial Version:

Referee #1 (Remarks to the Author):

In the present manuscript authors demonstrate the successful synthesis of a new member of cyclocarbon family containing 16 Carbon atoms and having doubly antiaromatic character. Principally, the approach utilized in this report to synthesize cyclo[16]carbon is very similar to that was used by authors in their previous publications (Ref. 10 and Ref. 21). In ref. 10 authors used the C18(CO)₆ precursor to synthesize C18 on NaCl surface and in Ref. 21 they used another C18Br₆ precursor to synthesize C18 on NaCl surface with improved product yield comparing to Ref. 10. In the current MS authors still use the same methodology with only difference that they proceed with mixed precursor C18Br₂(CO)₄. Thus, the methodological and conceptual novelty of the presented MS is very limited. The idea that C16 is doubly antiaromatic species is not principally new because of D_{8h} C16 was predicted to be antiaromatic computationally in some previous reports. Additionally, the fact itself that authors synthesized and characterized doubly antiaromatic C_{4n} ring that should be extremely reactive and unstable in respect to the rearrangement or transformation into the aromatic/nonaromatic system is also expected at experimental conditions realized by authors (T=5K, tip-induced fixation of the molecule on NaCl surface which prevent out-of-plane distortions of C16). There are however clear aspects of novelty incl. characterization of charged (C16)⁻ and (C16)²⁻ species, solid theoretical characterization of different C16 isomers (paragraph before the conclusions section) and strong chemical part leading to precursor 7.

I should note that following the statement in the introduction part that |2200> ground state of the D_{16h} C16 with no BLA should be doubly aromatic system with 18 (4n + 2) and 14 (4n - 2) electrons in out-of-plane and in-plane π-systems, I, however, missed this point in the end of discussion section. I recommend authors to update theoretical part with aromaticity indices for no-BLA |2200> and |1111> states of C16.

Finally, accounting for the limited conceptual novelty of the present manuscript, I unfortunately can not recommend to publish it in the Nature journal, while Nat. Commun. could be a good platform for publishing this interesting study.

Referee #2 (Remarks to the Author):

The Manuscript On-Surface Synthesis of a Doubly Anti-Aromatic 1 Carbon Allotrope: Cyclo[16]carbon by Yueze Gao, Florian Albrecht, Igor Rončević, Isaac Ettetdgui, Paramveer Kumar, Lorel M. Scriven, Kirsten E. Christensen, Shantanu Mishra, Luca Righetti, Max Rossmannek, Ivano Tavernelli, Harry L. Anderson and Leo Gross describes an on-surface synthesis of cyclo[16]carbon (C16) from the corresponding dehydroannulene derivative, characterization of the title compound, and theoretical support of some experimental results. The manuscript is well written and comprehensively describes

this important achievement of the first structural characterization of antiaromatic cyclocarbon. While I have no doubts about the importance of this achievement, I have some comments to the manuscript:

1. The description of the failed synthetic pathway is distracting and off-topic, and it detracts from the main achievement of the study. While it is worth briefly mentioning this in the main manuscript, a paragraph and a half of a figure in Nature, which has a broad readership, is excessive. Most of this information should be moved to the Supplementary Information (SI) to maintain the focus on the actual achievement. Additionally, it is surprising that the authors describe the synthesis of known molecules (3A, 3C, 4C) using a numbering system that is not even mentioned in the main text. This information should also be included in the SI, rather than the main manuscript, to avoid further detracting from the main achievement of the study.

2. The discussion in the paper appears to place an excessive emphasis on the theoretical description of C16 and its charged states, while the experimental synthesis and characterization of the compound is a significant achievement in its own right. While theoretical data is important to support experiments, the balance of the discussion seems to have shifted too much towards the theoretical side. There have been numerous theoretical papers published on cyclocarbons, and therefore, the discussion of, for example, the aromaticity of C18 and C16 may not be novel enough for inclusion in the main text of a Nature paper. It may be more appropriate to focus on the unique experimental features and findings of the study, while briefly mentioning the theoretical underpinnings that support the experimental results.

3. The authors mention in the caption to Figure S13 that C16 was never observed stably on the NaCl surface, but always near the 3rd NaCl layer, and that it has a shallow adsorption energy potential landscape. These are important points that should be discussed in the main text, as they shed light on the underlying mechanism of C16 synthesis and characterization. In the caption to Figure S15 and Table S2, the authors also postulate the impact of the 3rd NaCl layer on the adsorption. This hypothesis should be further explored and discussed in the main text, given the broad expertise in computational chemistry demonstrated by the authors. It would be worthwhile to calculate the C16 adsorption on the NaCl energy potential landscape and the impact of the 3rd NaCl layer adjacent to the C16 molecule, as this appears to be a key factor in the success of the synthesis approach. By discussing these important findings and exploring their underlying mechanisms, the authors can provide a more comprehensive understanding of the synthesis and characterization of C16. In summary, the synthesis and characterization of C16 is a significant addition to the cyclocarbon family and is of broad interest as new carbon allotropes always are. Additionally, this research will contribute to the ongoing discussion on the concept of aromaticity. However, it is important to emphasize the actual findings of the study rather than repeating already reported theoretical works. To further advance the understanding of the synthesis and characterization of C16, it would be beneficial to conduct a theoretical investigation on the step edge of the 3rd layer NaCl. After including these corrections, I will be happy to recommend this important article for publication in Nature.

Referee #3 (Remarks to the Author):

This is the review report for the manuscript (MS) "On-Surface Synthesis of a Doubly Anti-Aromatic Carbon Allotrope: Cyclo[16]carbon" by Gao et al. The MS reports on the synthesis of a cyclic C16 molecule by scanning probe microscopy tip-induced atom manipulation at low temperature on a NaCl surface. The molecule's structural and electronic properties are investigated by scanning tunneling and atomic force microscopy (STM/AFM) and various methods of electronic structure calculations.

The study's main findings are the successful synthesis of C16 in condensed phase, which had not been achieved previously, presumably due to its high reactivity caused by the double anti-aromatic character. AFM images show significant bond length alternation (BLA), confirming its predicted polyynic structure. On the NaCl surface, the molecule can be switched between the neutral and anionic charge state, and the orbital resonance corresponding to electron injection (LUMO) into anionic C16 was imaged. Comparison with the calculated orbitals together with the observed BLA confirm the doubly anti-aromatic character of the C16.

The MS is very well written, in concise language and well-structured. The data is of high quality, very extensive and supports the conclusions of the paper.

The synthesis of cyclocarbons in condensed phase proved extremely challenging for decades, and its first successful demonstration in 2019 (for cyclo[18]carbon, by some of the authors of the present MS) had a large impact, in particular in the field of chemistry, and led to a string of theoretical papers investigating the properties and potential applications of cyclocarbons. However, synthesis of other cyclocarbons had not been reported until now. This MS, and a recent preprint demonstrating the on-surface synthesis of cyclo[10]carbon and cyclo[14]carbon (ref. 9 in the present MS) are the first reports on the successful synthesis of other cyclocarbons. C16 differs from C10, C14, and C18 in that it should be anti-aromatic (while the others are aromatic), and this is convincingly demonstrated in the MS. The precursor molecule for the on-surface synthesis is similar to the ones employed for C18 (it can be thought of as a combination of the two), suggesting that this precursor family might also enable the on-surface synthesis of other cyclocarbons. The MS also provides the first real space image of one of cyclocarbon's molecular orbitals, which remained elusive for C10, C14, and C18. Thus, I think this MS should be of large interest to the chemistry community and the field of on-surface synthesis and is suitable for publication in Nature.

However, there are a couple of technical comments that I would like the authors to address first:

Major:

#1: l. 109 ff.: The authors give quite broad voltage ranges for debromination and removal of CO masking groups. Could the authors elaborate a bit more on those values? Do they depend on the tunneling current? Does the electric field in the junction play role? Is the dissociation mechanism the same for all those voltages or are different mechanism at play depending on the bias?

#2: Regarding the oval shape of the anionic C16 on NaCl: Do I understand correctly that the oval

shape is a result of the particular adsorption geometry of the charged molecule and not an intrinsic property of the charged molecule?

#3: Regarding the charge state switching: Was this possible for all/most generated C16 molecules only for the one shown in Fig. 4, presumably due to its particular adsorption geometry? I did not find a clear statement on this in the MS.

#4: l. 186 ff. I find the description of the theoretical results concerning the ring current and NICS values not very accessible for non-experts. Considering the broad readership of Nature, would it be possible to add a short explanation of their physical meaning and significance?

#5: SM, Fig. S6 l: In the dI/dV measurements, there is a peak at around -0.3 V, before the resonance shown in the STM in panel (j). What is the origin of this peak?

#6: SM, Figs. S16 and S17b: The KPFS data showing the charge state switching seems to be incomplete. For the current trace, the complete forward and backward sweeps is shown, but to me it looks like this is not the case of the KPFS trace. At least it looks like the points at the actual transition are missing, and also the part of the backward sweep after switching back. If this should indeed be the case, please show the full sweep of the KPFS trace in Fig. S16 and S17b.

Minor:

#7: l. 86: "Fortunately, we were able to get round this problem [...]" This formulation sounds a bit colloquial, maybe the authors can reformulate this.

#8: l. 106: I think here is missing an opening parenthesis (or there is a closing one too much).

#9: SM, p. 11, section Additional STM and AFM data: "[...], applying voltage pulses the location of an individual precursor molecule" There might be missing `at` after "pulses"?

#10: SM, caption Fig. S5: The authors refer to the AFM image recorded with constant-current feedback as "dynamic AFM". I would advise against using this term here. Historically, dynamic AFM referred to AFM modes where the cantilever is deliberately oscillated, i.e., tapping mode AFM and noncontact AFM; in contrast to static AFM where the tip does not oscillate, i.e., contact AFM.

#11: SM, Fig. S14 b: The AFM image is missing the grid visualizing the NaCl lattice. Is this on purpose? If so, why?

Author Rebuttals to Initial Comments:

Reply to Points from Reviewer 1

Comment: In the present manuscript authors demonstrate the successful synthesis of a new member of cyclocarbon family containing 16 Carbon atoms and having doubly antiaromatic character. Principally, the approach utilized in this report to synthesize cyclo[16]carbon is very similar to that was used by authors in their previous publications (Ref. 10 and Ref. 21). In ref. 10 authors used the C18(CO)₆ precursor to synthesize C18 on NaCl surface and in Ref. 21 they used another C18Br₆ precursor to synthesize C18 on NaCl surface with improved product yield comparing to Ref. 10. In the current MS authors still use the same methodology with only difference that they proceed with mixed precursor C18Br₂(CO)₄. Thus, the methodological and conceptual novelty of the presented MS is very limited. The idea that C16 is doubly antiaromatic species is not principally new because of D_{8h} C16 was predicted to be antiaromatic computationally in some previous reports. Additionally, the fact itself that authors synthesized and characterized doubly antiaromatic C_{4n} ring that should be extremely reactive and unstable in respect to the rearrangement or transformation into the aromatic/nonaromatic system is also expected at experimental conditions realized by authors (T=5K, tip-induced fixation of the molecule on NaCl surface which prevent out-of-plane distortions of C16).

There are however clear aspects of novelty incl. characterization of charged (C16)⁻ and (C16)²⁻ species, solid theoretical characterization of different C16 isomers (paragraph before the conclusions section) and strong chemical part leading to precursor 7.

Response: We thank the reviewer for their comments. C₁₆ is the first doubly anti-aromatic molecule of any type to be structurally characterized which makes this work important. The ability to switch C₁₆ between different charge states, and to characterise those charge states on the surface, is also remarkable. Although the synthetic strategy presented in this paper builds on our previous publications, the use of cross-coupling reactions to make an unsymmetrical precursor, with both C2 and C4 links, and with both types of masking groups, takes this chemistry to a new level. Doubly antiaromatic C₁₆ is expected to be even less stable than the previously demonstrated doubly aromatic C₁₈. Thus, it was not obvious that C₁₆ could be generated and characterized.

Comment: I should note that following the statement in the introduction part that |2200> ground state of the D_{16h} C16 with no BLA should be doubly aromatic system with 18 (4n + 2) and 14 (4n - 2) electrons in out-of-plane and in-plane π-systems, I, however, missed this point in the end of discussion section. I recommend authors to update theoretical part with aromaticity indices for no-BLA |2200> and |1111> states of C16.

Response: We thank the referee for this point. We have now included NICS plots to provide aromaticity indices for both the |2200> and |2020> configurations of C₁₆ in the Supplementary Information. (Analysing the aromaticity of the |1111> configurations of C₁₆ at zero-BLA is more complicated because it is an open-shell configuration. In first-order perturbation theory, the NICS values of |1111> at no-BLA are not well-defined, which we also mention in the caption of Fig. S28.)

Reply to Points from Reviewer 2

Comment: The Manuscript On-Surface Synthesis of a Doubly Anti-Aromatic 1 Carbon Allotrope: Cyclo[16]carbon by Yuezhe Gao, Florian Albrecht, Igor Rončević, Isaac Etedgui, Paramveer Kumar, Lorel M. Scriven, Kirsten E. Christensen, Shantanu Mishra, Luca Righetti, Max Rossmannek, Ivano

Tavernelli, Harry L. Anderson and Leo Gross describes an on-surface synthesis of cyclo[16]carbon (C₁₆) from the corresponding dehydroannulene derivative, characterization of the title compound, and theoretical support of some experimental results. The manuscript is well written and comprehensively describes this important achievement of the first structural characterization of antiaromatic cyclocarbon.

Response: We thank the reviewer for their positive evaluation.

Comment: 1. The description of the failed synthetic pathway is distracting and off-topic, and it detracts from the main achievement of the study. While it is worth briefly mentioning this in the main manuscript, a paragraph and a half of a figure in Nature, which has a broad readership, is excessive. Most of this information should be moved to the Supplementary Information (SI) to maintain the focus on the actual achievement. Additionally, it is surprising that the authors describe the synthesis of known molecules (3A, 3C, 4C) using a numbering system that is not even mentioned in the main text. This information should also be included in the SI, rather than the main manuscript, to avoid further detracting from the main achievement of the study.

Response: As suggested, we have now moved the description of the unsuccessful route to C₁₆ to the Supplementary Information. We no longer mention any compounds in the main text without showing their structural formulae in the main text.

Comment: 2. The discussion in the paper appears to place an excessive emphasis on the theoretical description of C₁₆ and its charged states, while the experimental synthesis and characterization of the compound is a significant achievement in its own right. While theoretical data is important to support experiments, the balance of the discussion seems to have shifted too much towards the theoretical side. There have been numerous theoretical papers published on cyclocarbons, and therefore, the discussion of, for example, the aromaticity of C₁₈ and C₁₆ may not be novel enough for inclusion in the main text of a Nature paper. It may be more appropriate to focus on the unique experimental features and findings of the study, while briefly mentioning the theoretical underpinnings that support the experimental results.

Response: We have reworded the discussion to emphasize how the antiaromaticity of C₁₆ relates to its electronic structure and orbitals, and how the densities of these orbitals are observed experimentally. We have also moved the (extended) theoretical NICS results into the SI.

Comment: 3. The authors mention in the caption to Figure S13 that C₁₆ was never observed stably on the NaCl surface, but always near the 3rd NaCl layer, and that it has a shallow adsorption energy potential landscape. These are important points that should be discussed in the main text, as they shed light on the underlying mechanism of C₁₆ synthesis and characterization. In the caption to Figure S15 and Table S2, the authors also postulate the impact of the 3rd NaCl layer on the adsorption. This hypothesis should be further explored and discussed in the main text, given the broad expertise in computational chemistry demonstrated by the authors. It would be worthwhile to calculate the C₁₆ adsorption on the NaCl energy potential landscape and the impact of the 3rd NaCl layer adjacent to the C₁₆ molecule, as this appears to be a key factor in the success of the synthesis approach. By discussing these important findings and exploring their underlying mechanisms, the authors can provide a more comprehensive understanding of the synthesis and characterization of C₁₆.

Response:

As suggested by the reviewer, we have modified page 5 of the main text to read "We never observed neutral C₁₆ stably isolated on the NaCl surface, but always near a 3rd layer NaCl step edge, to provide a more stable adsorption site and facilitate detailed characterisation."

We computationally investigated the interactions of neutral and negatively charged C₁₆ with six distinct defects on a NaCl surface, with the defect structure inspired by the AFM images. These results have been added to Table S1, while the adsorption energies and changes in cyclocarbon shape are summarised in Figure S54. The following two sentences have been added to the main text (page 5):

"Our computations (Table S1 and Figs. S54 and S55) predict a larger adsorption energy (1.13 and 2.61 eV for C₁₆ and C₁₆⁻, respectively) at defect sites compared to the pristine surface, accompanied by an increase in BAA (up to 35° for C₁₆⁰ and up to 50° for C₁₆⁻, compared to 20–30° on a pristine surface). BLA is maintained in all cases, with no fundamental changes in the electronic structure."

Comment: In summary, the synthesis and characterization of C16 is a significant addition to the cyclocarbon family and is of broad interest as new carbon allotropes always are. Additionally, this research will contribute to the ongoing discussion on the concept of aromaticity. However, it is important to emphasize the actual findings of the study rather than repeating already reported theoretical works. To further advance the understanding of the synthesis and characterization of C16, it would be beneficial to conduct a theoretical investigation on the step edge of the 3rd layer NaCl. After including these corrections, I will be happy to recommend this important article for publication in Nature.

Response: We thank the reviewer for a positive evaluation. NICS plots have been moved to the SI (Fig. S28), and the results of our theoretical investigation of C₁₆ on step edges are now discussed in the main text (and reported in-depth in Figs. S54, S55 and Table S1).

Reply to Points from Reviewer 3

Comment: This is the review report for the manuscript (MS) "On-Surface Synthesis of a Doubly Anti-Aromatic Carbon Allotrope: Cyclo[16]carbon" by Gao et al. The MS reports on the synthesis of a cyclic C16 molecule by scanning probe microscopy tip-induced atom manipulation at low temperature on a NaCl surface. The molecule's structural and electronic properties are investigated by scanning tunneling and atomic force microscopy (STM/AFM) and various methods of electronic structure calculations.

The study's main findings are the successful synthesis of C16 in condensed phase, which had not been achieved previously, presumably due to its high reactivity caused by the double anti-aromatic character. AFM images show significant bond length alternation (BLA), confirming its predicted polyynic structure. On the NaCl surface, the molecule can be switched between the neutral and anionic charge state, and the orbital resonance corresponding to electron injection (LUMO) into anionic C16 was imaged. Comparison with the calculated orbitals together with the observed BLA confirm the doubly anti-aromatic character of the C16.

The MS is very well written, in concise language and well-structured. The data is of high quality, very extensive and supports the conclusions of the paper.

The synthesis of cyclocarbons in condensed phase proved extremely challenging for decades, and its first successful demonstration in 2019 (for cyclo[18]carbon, by some of the authors of the present MS) had a large impact, in particular in the field of chemistry, and led to a string of theoretical papers investigating the properties and potential applications of cyclocarbons. However, synthesis of other cyclocarbons had not been reported until now. This MS, and a recent preprint demonstrating the on-surface synthesis of cyclo[10]carbon and cyclo[14]carbon (ref. 9 in the present MS) are the first reports on the successful synthesis of other cyclocarbons. C16 differs from C10, C14, and C18 in that it should be anti-aromatic (while the others are aromatic), and this is convincingly demonstrated in the MS. The precursor molecule for the on-surface synthesis is similar to the ones employed for C18 (it can be thought of as a combination of the two), suggesting that this precursor family might also enable the on-surface synthesis of other cyclocarbons. The MS also provides the first real space image of one of cyclocarbon's molecular orbitals, which remained elusive for C10, C14, and C18. Thus, I think this MS should be of large interest to the chemistry community and the field of on-surface synthesis and is suitable for publication in Nature.

Response: We thank the reviewer for this enthusiastic evaluation.

Comment: #1: l. 109 ff.: The authors give quite broad voltage ranges for debromination and removal of CO masking groups. Could the authors elaborate a bit more on those values? Do they depend on the tunneling current? Does the electric field in the junction play role? Is the dissociation mechanism the same for all those voltages or are different mechanism at play depending on the bias?

Response: The voltage onset of 1.3 V for debromination corresponds to the onset of first ionic resonance at positive bias of **4** on the surface (see Fig. S6, panel o). At this bias, electrons are attached transiently from the tip to the molecule, thus they change the charge state of **4** (to dianionic), and importantly they can transfer energy (that might result in the dissociation reaction) by inelastic electron tunnelling (IET) processes, see ref. [32], i.e., Pavlicek et al. *Nat. Chem.* 10, 853–858 (2018), Fig. S29. For CO unmasking, typically larger voltages were required. We performed the voltage pulses at constant tip height. We started with a large tip height resulting in tunnelling currents below the detection

threshold (about 10 fA). After each pulse, we checked whether the pulse had an effect by recording STM and/or AFM images. For different voltages we successively decreased the tip height of pulses until the pulse resulted in a reaction. Typically, dissociation reactions occurred when reaching currents on the order of few pA (at voltages of 1.3 V for Br dissociation and at about 3 V for CO dissociation). Due to the challenging experiment (many different intermediates, different dissociation reactions, dislocation of molecules when applying voltage pulses), a detailed systematic study of the reaction mechanisms is out of the scope of this paper. We previously performed such a systematic study, i.e., dependence on voltage and tunnel current, for debromination reactions (ref. 32) and hypothesise that here the mechanisms are similar. Addressing the dependence on the tunnel current: In a single-electron IET process, the tunnel current is proportional to the reaction rate. Note that in that previous paper and for the reaction investigated there, we found single-electron yields on the order of 10^{-9} , that means that a typical time to trigger a reaction is on the order of 10 s for currents on the order of 10 pA, i.e., on the order of 10^{-8} electrons/second, and current and rate are proportional to each other. Also, here we observe that the likeliness of a reaction occurring during a pulse of a few seconds increases with increasing tunnelling current (at fixed bias). The single electron yields would be similar in our current experiment, where we observe reactions at few pA current occurring on the order of several seconds. Few CO dissociations events occurred at lower bias, down to 1.5 V, possibly those are triggered by a two-electron process, (see Y. Kim, et al. *Phys. Rev. Lett.* 89, 126104 (2002)), but this is speculative. We do not have sufficient data for a statistical analysis as performed in ref. 32, and due to the challenging experiment, to collect such data for this system is not feasible.

We have revised the text on page 4 to read:

“Voltage pulses, applied for a few seconds at constant tip height, were used to unmask individual precursors **4**. We successively increased the voltage and decreased the tip height for the pulse until it resulted in dissociation reactions. For tunnelling currents on the order of few pA, the minimum voltage required for debromination of **4** to give **5** (Fig 3c, and Fig. S7 for additional data) was 1.3 V, coinciding with the bias for resonant tunnelling, i.e., electron attachment to **4** (see Fig. S6). For CO unmasking, larger bias voltages were required, typically about 3 V. We speculate that the dissociation reactions are triggered in transiently charged species by inelastic electron tunnelling processes³².”

In this we compare biases for similar currents. We deleted the sentence with the wider voltage ranges that was previously in the caption of Fig. 3.

Comment: #2: Regarding the oval shape of the anionic C₁₆ on NaCl: Do I understand correctly that the oval shape is a result of the particular adsorption geometry of the charged molecule and not an intrinsic property of the charged molecule?

Response: Yes, the oval shape is the result of interactions of anionic C₁₆ with the surface. It also reflects the fact that this anion is easily deformed. The anion is more flexible than neutral C₁₆ (see Fig. S30) and it interacts more strongly with the NaCl surface (Table S1). In addition, in our added calculations the effect of the third layer step on the shape can be observed (Fig. S54c). Again, more significant deformations are observed for the anions than for the neutral molecule.

Comment: #3: Regarding the charge state switching: Was this possible for all/most generated C₁₆ molecules only for the one shown in Fig. 4, presumably due to its particular adsorption geometry? I did not find a clear statement on this in the MS.

Response: Yes, it is possible for most generated C₁₆. However, sometimes the molecule changed adsorption site when we ramped the bias and thus these molecules were only characterized at one charge state at this site. We added the following text to the caption of Fig. S13, which show molecules that charge state and adsorption site were switched:

“Typically, that is, at most adsorption sites, we observed charge state bistability of neutral and anionic charge states of C₁₆. In some cases, the molecule changed its adsorption site when the voltage was ramped up to switch the charge state. The molecule in Fig. 4 did not move when the charge state was switched repeatedly and for many times, nor when the orbital density was mapped, presumably due to a very stable adsorption site at a 3rd layer NaCl island (bay 1, see Fig. S55 and Table S1).”

Comment: #4: l. 186 ff. I find the description of the theoretical results concerning the ring current and NICS values not very accessible for non-experts. Considering the broad readership of Nature, would it be possible to add a short explanation of their physical meaning and significance?

Response: We have added a short explanation of the ring currents and moved the discussion of the NICS to the SI.

Comment: #5: SM, Fig. S6 I: In the dI/dV measurements, there is a peak at around -0.3 V, before the resonance shown in the STM in panel (j). What is the origin of this peak?

Response: This feature is related to the onset of the interface state. This state at the interface of Cu(111) and NaCl descends from the Shockley surface state of Cu(111), which survives as an interface state upon coverage of Cu(111) with NaCl [J. Repp et al. *Phys. Rev. Lett.* 92, 36803 (2004)]. In our paper, we employ the interface state to infer about the charge states of C_{16} , as only charged species lead to interface state scattering [J. Repp, et al. *Science.* 305, 493–495 (2004)] (see Fig. S12).

We have added the following sentence to the caption of Figure S7: “The feature at -0.3 V relates to the onset of the Cu(111)/NaCl interface state³².”

Comment: #6: SM, Figs. S16 and S17b: The KPFS data showing the charge state switching seems to be incomplete. For the current trace, the complete forward and backward sweeps is shown, but to me it looks like this is not the case of the KPFS trace. At least it looks like the points at the actual transition are missing, and also the part of the backward sweep after switching back. If this should indeed be the case, please show the full sweep of the KPFS trace in Fig. S16 and S17b.

Response: We added additional panels with the raw data of the same KPFM measurements for the full sweep range, in which the switching events can be observed (Fig. S17b,c and Fig. S18c,d). The graphs in the former images (now panels Fig S17a and Fig. S18b, are low-pass filtered to better visualize the parabolas and to access their shifted peak positions in one plot. That is, to have the figure readable and to not have the graphs (with the raw data noise) overlapping. In these low-pass filtered graphs we only use data until a charge transition occurs, that is data points corresponding to only one parabola for each sweep direction. We keep the original low-pass filtered graphs and describe the low-pass filtering in the caption and in addition show the raw data.

Action: New panels have been added showing the corresponding raw KPFM data and expanded captions in Figures S17 and S18.

Comment: #7: I. 86: “Fortunately, we were able to get round this problem [...]” This formulation sounds a bit colloquial, maybe the authors can reformulate this.

Response: This sentence has now been removed.

Comment: #8: I. 106: I think here is missing an opening parenthesis (or there is a closing one too much).

Response: We thank the reviewer for noticing this error. It has now been corrected.

Comment: #9: SM, p. 11, section Additional STM and AFM data: “[...], applying voltage pulses the location of an individual precursor molecule” There might be missing ‘at’ after ‘pulses’?

Response: We thank the referee for guiding us to this mistake. As suggested, we added the “at” after “pulses”.

Comment: #10: SM, caption Fig. S5: The authors refer to the AFM image recorded with constant-current feedback as “dynamic AFM”. I would advise against using this term here. Historically, dynamic AFM referred to AFM modes where the cantilever is deliberately oscillated, i.e., tapping mode AFM and noncontact AFM; in contrast to static AFM where the tip does not oscillate, i.e., contact AFM.

Response: We deleted the term “dynamic AFM”, which was indeed misleading, and provide the technical details of this acquisition mode, only.

Comment: #11: SM, Fig. S14 b: The AFM image is missing the grid visualizing the NaCl lattice. Is this on purpose? If so, why?

Response: We thank the referee for spotting this omission. We added the grid visualization.

Reviewer Reports on the First Revision:

Referee #1 (Remarks to the Author):

I can accept argumentation by Authors regarding the novelty of the paper and generally the revised manuscript looks better comparing to the initial version. Despite I still a bit skeptic about the novelty of the approach leading to C16, I can agree that the fact itself that doubly antiaromatic cyclocarbon was structurally characterized for the first time, makes this report suitable for publication in Nature.

Referee #2 (Remarks to the Author):

I stand by my opinion that this manuscript reports a significant discovery, which is of interest to the broad readership of Nature. It is well written and the data presented is solid. The authors addressed all of my comments to the first version of the manuscript. It reads very well with a focus on the main discoveries. I have several minor comments:

P3L74: I think Fig. S3 should be Fig. S2 as Fig. S2 shows NMR comparison

P4L82: "partially covered with bilayer NaCl" – this might be confusing because the step-edge to the 3rd layer is discussed later. I would suggest rephrasing slightly.

P4L90: 'were used to unmask individual precursors 4' should be: 'were used to unmask acetylenes from individual precursors 4'

P8L204: The last sentence of the conclusions paragraph seems to be almost the same as the last sentence of the abstract.

SI

- Are the red references intentional?

After correcting these minor issues I am happy to recommend publication in Nature.

Referee #3 (Remarks to the Author):

The authors addressed all my comments in satisfactory manner and made corresponding changes to the manuscript and supplementary information. I also think that the DFT calculations of the adsorption position of C16 next to third layer NaCl (in response to reviewer #2) adds valuable information to the manuscript. I still found a few typos (see below), but otherwise recommend publication of the manuscript in Nature.

#1: Figure 3 caption (l. 135): "a reaction scheme" to be consistent with the rest of the caption, it probably should be "a, Reaction scheme."

#2: SI, p. 3: "PBE (12)" here the format of the reference is different from the other ones (which are superscripts).

#3: SI, Figure S17 caption: "(c, d)" should be "(b, c)" I guess.

Author Rebuttals to First Revision:

Nature manuscript 2023-04-06260A

Title: *On-Surface Synthesis of a Doubly Anti-Aromatic Carbon Allotrope*

Reply to Reviewer #1

I can accept argumentation by Authors regarding the novelty of the paper and generally the revised manuscript looks better comparing to the initial version. Despite I still a bit skeptic about the novelty of the approach leading to C16, I can agree that the fact itself that doubly antiaromatic cyclocarbon was structurally characterized for the first time, makes this report suitable for publication in Nature..

Response: We thank the referee for their positive evaluation of the manuscript.

Reply to Reviewer #2

I stand by my opinion that this manuscript reports a significant discovery, which is of interest to the broad readership of Nature. It is well written and the data presented is solid. The authors addressed all of my comments to the first version of the manuscript. It reads very well with a focus on the main discoveries. I have several minor comments:

(1) P3L74: I think Fig. S3 should be Fig. S2 as Fig. S2 shows NMR comparison.

Response: We thank the reviewer for noticing this point. It has now been corrected.

(2) P4L82: "partially covered with bilayer NaCl" – this might be confusing because the step-edge to the 3rd layer is discussed later. I would suggest rephrasing slightly.

Response: We have slightly reworded these three sentences to clarify this point. The text now reads: "Precursor **4** was sublimed by fast heating from a Si wafer¹⁰ onto a Cu(111) single crystal surface partially covered with NaCl at a sample temperature of about $T = 10$ K. On-surface synthesis (Fig. 3a) and characterisation by STM and AFM with CO tip functionalisation^{30,32} were performed at $T = 5$ K. We found intact molecules of **4** on bilayer NaCl, denoted NaCl(2ML)/Cu(111), as shown in Fig. 3b.

(3) P4L90: 'were used to unmask individual precursors **4**' should be: 'were used to unmask acetylenes from individual precursors **4**'.

Response: We have changed "were used to unmask individual precursors **4**." to "were used to unmask the acetylenes in individual molecules of precursor **4**."

(4) P8L204: The last sentence of the conclusions paragraph seems to be almost the same as the last sentence of the abstract.

Response: We have deleted the last sentence of the conclusions to remove this duplication.

(5) SI - Are the red references intentional?

Response: We have changes al the text to black.

Reply to Reviewer #3

The authors addressed all my comments in satisfactory manner and made corresponding changes to the manuscript and supplementary information. I also think that the DFT calculations of the adsorption position of C16 next to third layer NaCl (in response to reviewer #2) adds valuable information to the manuscript. I still found a few typos (see below), but otherwise recommend publication of the manuscript in Nature.

(1) Figure 3 caption (l. 135): "a reaction scheme" to be consistent with the rest of the caption, it probably should be "a, Reaction scheme."

Response: We agree and have changed the punctuation as suggested.

(2) SI, p. 3: "PBE (12)" here the format of the reference is different from the other ones (which are superscripts).

Response: We thank the referee for noticing this point. We have now made reference number 12 superscript to be consistent.

(3) SI, Figure S17 caption: "(c, d)" should be "(b, c)" I guess.

Response: Yes. We have now corrected this point.